A flexible perception method of thin smoke based on patch total bounded variation for buildings

Zhang Jieming zjieming1206@163.com
Gao Yifan
Chen Xianchao
Chen Zhanchen
Zhaoqing Power Supply Bureau of Guangdong Power Grid Co., Ltd. , Zhaoqing, Guangdong , China
Alatas Bilal
Electronic publication date: 2024 Sep 19
Publication date: 2024
Volume: 10
Electronic Location ID: e2282
Received 2024 Mar 19; Accepted 2024 Aug 5
Copyright: © 2024 Zhang et al.
Copyright year: 2024
Copyright holder: Zhang et al.
License: This is an open access article distributed under the terms of the Creative Commons Attribution License, which permits unrestricted use, distribution, reproduction and adaptation in any medium and for any purpose provided that it is properly attributed. For attribution, the original author(s), title, publication source (PeerJ Computer Science) and either DOI or URL of the article must be cited.
License URL: https://creativecommons.org/licenses/by/4.0/

Keywords: Total bounded variation, Video smoke detection, Micro-amplification, The image energy ratio

Funding: Science and technology project of China Southern Power Grid Company Limited 031200KK52200003 This article was funded by the science and technology project of China Southern Power Grid Company Limited (031200KK52200003). The funders had no role in study design, data collection and analysis, decision to publish, or preparation of the manuscript.

==============================
Early fire warning is critical to the safety and stability of power systems. However, current methods encounter challenges in capturing subtle features, limiting their effectiveness in providing timely alerts for potential fire hazards. To overcome this drawback, a novel detection algorithm for thin smoke was proposed to enhance early fire detection capabilities. The core is that the Patch-TBV feature was proposed first, and the total bounded variation (TBV) was computed at the patch level. This approach is rooted in the understanding that traditional methods struggle to detect minute variations in image characteristics, particularly in scenarios where the features are dispersed or subtle. By computing TBV at a more localized level, the algorithm proposed gains a finer granularity in assessing image quality, enabling it to capture subtle variations that might indicate the presence of smoke or early signs of a fire. Another key aspect that sets our algorithm apart is the incorporation of subtle variation magnification. This technique serves to magnify subtle features within the image, leveraging the computed TBV values. This magnification strategy is pivotal for improving the algorithm’s precision in detecting subtle variations, especially in environments where smoke concentrations may be minimal or dispersed. To evaluate the algorithm’s performance in real-world scenarios, a comprehensive dataset, named TIP, comprising 3,120 images was constructed. The dataset covers diverse conditions and potential challenges that might be encountered in practical applications. Experimental results confirm the robustness and effectiveness of the proposed algorithm, showcasing its ability to provide accurate and timely fire warnings in various contexts. In conclusion, our research not only identifies the limitations of existing methods in capturing subtle features for early fire detection but also proposes a sophisticated algorithm, integrating Patch-TBV and micro-variation amplification, to address these challenges. The algorithm’s effectiveness and robustness are substantiated through extensive testing, demonstrating its potential as a valuable tool for enhancing fire safety in power systems and similar environments.

Introduction

In certain industries, maintaining uninterrupted server operations is crucial, but this necessity also raises concerns about the potential for fires due to various reasons. Recent incidents, such as the SK Corporation incident in South Korea and the OVH fire in France, underscore the urgent requirement for cutting-edge technologies that enhance early detection capabilities. These occurrences highlight the significance of implementing state-of-the-art fire detection systems in server rooms and data centers, ensuring the resilience and security of critical infrastructure as the reliance on uninterrupted server operations continues to grow.

However, traditional Early Fire Warning systems based on smoke sensors have limitations, including a small detection range and the necessity for specific changes in physical quantities such as smoke, gas, and temperature to trigger an alarm. As a result, they may not be suitable for providing early warnings. With the advancement of visual technology and corresponding improvements in monitoring equipment, utilizing video images collected by monitoring systems for fire warnings can effectively issue timely early warnings. Therefore, the future application prospects for vision-based fire detection and early warning methods appear promising.

Recent advancements in smoke detection have integrated various innovative methodologies and deep learning techniques to enhance detection capabilities:

1) Video Processing and Image Analysis: Several approaches have utilized video processing techniques for early fire and smoke detection. Methods such as background subtraction, color analysis, and motion estimation are employed to identify smoke and flames effectively (Qureshi et al., 2016). Convolutional neural networks (CNNs) have been used to analyze video streams, demonstrating the power of deep learning in enhancing detection accuracy (Frizzi et al., 2016). Additionally, machine vision and image processing techniques have been applied to recognize smoke patterns (Yuanbin, 2016), and color combined with wavelet analysis has been used to detect both smoke and flame (Shiping et al., 2017).

2) Deep Learning Enhancements: Deep learning algorithms have been refined to improve detection performance. Techniques like YOLOv3 with attention mechanisms and anchor-free methods have been utilized to enhance feature extraction and reduce false alarms (Sun & Feng, 2023). The RepVGG-YOLOv7 model incorporates ECA attention mechanisms and SIoU loss functions to improve detection accuracy, especially for small smoke targets (Chen et al., 2023). Lightweight models such as improved YOLOv5 have been proposed for efficient early forest fire detection (Chen et al., 2024), and global awareness deep network models combine path aggregation networks with YOLOv3 variants to optimize real-time detection (Xiao & Yan, 2023).

3) Integration with Modern Technologies: Modern technologies have been integrated into smoke detection systems to enhance their effectiveness. IoT and smart technologies support advanced detection systems, such as integrated fire detection for smart cities and intelligent smoke alarms using wireless networks and thermal infrared sensing (Huang et al., 2023; Zhan et al., 2022; Peng & Wang, 2019; Hu et al., 2022; Chaturvedi, Khanna & Ojha, 2022). Methods involving image fusion, blind image restoration, and data fusion have also been explored to address challenges in smoke detection and improve system performance (Cheng et al., 2013; Namozov & Im Cho, 2018; Torabnezhad, Aghagolzadeh & Seyedarabi, 2013).

Overall, the combination of advanced image processing, deep learning techniques, and modern technological integrations significantly improves the accuracy and efficiency of smoke detection systems.

Currently, most smoke detection algorithms (Bahhar et al., 2023; Wang et al., 2022) are primarily used for detecting forest fires, aimed at enhancing the accuracy (Prema et al., 2022; Al-Smadi et al., 2023) and real-time response capabilities of these systems (Hu et al., 2022). These algorithms must adapt to the multi-scale nature of smoke (Huang et al., 2023) and complex environmental conditions (Zhan et al., 2022). However, compared to forest fire detection, indoor smoke detection faces challenges such as rapid dispersion in enclosed spaces and obstructions caused by indoor objects, which can hinder accurate determination of the smoke’s origin. Therefore, sensitive detection of early, minute signs of smoke is particularly crucial as it can help in promptly and effectively implementing firefighting measures.

Despite advancements in fire detection technologies, there remains a significant gap in detecting subtle smoke variations, especially in indoor power systems. Traditional methods struggle with early detection due to their reliance on physical changes like temperature or gas concentration. Our study addresses this gap by introducing the Patch-TBV feature, which enhances sensitivity to subtle smoke variations. This novel approach, combined with the TIP dataset, provides a robust solution for early fire detection, significantly improving upon existing methodologies. The main contributions of this article are as follows.

1) A dataset of thin smoke, named TIP (thin smoke image dataset for indoor power system), was constructed. A total of 3,120 images of thin smoke were adopted and different indoor scenarios and thin smoke levels were captured.

2) An image feature of thin smoke, patch-TBV, was proposed. The total bounded variation at the patch level was computed, which enhances sensitivity to subtle smoke variations.

3) A novel thin smoke detection algorithm based on patch-TBV was proposed. The subtle variation magnification and image energy ratio are combined with Patch-TBV. Therefore, the robustness and accuracy of the algorithm proposed was improved.

The rest of this article is organized as follows. In “Related Work”, related works about fire and smoke detection are introduced. In “Research Method”, the research method, including framework, TIP dataset, patch-TBV, and image energy ratio, are introduced. Based on it, the thin smoke detection algorithm was proposed. The data validation results and discussion presented in “Results and Discussions” underscore the superior performance of our proposed algorithm, showcasing its effectiveness through comprehensive comparisons with several existing methods. Finally, conclusions are given in “Conclusions”.

Related work

In recent years, smoke detection technology has emerged as a critical component of early fire detection and timely response mechanisms, garnering significant attention from both the academic and industrial sectors. With the rapid advancement of computer vision and deep learning, the methodologies and applications of smoke detection have become increasingly diverse and precise.

Qureshi et al. (2016) introduced the QuickBlaze system, which employs a video processing approach utilizing multiple techniques such as background subtraction, color analysis, and motion estimation for early fire detection. Frizzi et al. (2016) explored a convolutional neural network (CNN) for video-based fire and smoke detection, demonstrating the potent capabilities of deep learning in processing video streams. Yuanbin’s (2016) study delved into smoke recognition based on machine vision, applying image processing techniques for smoke detection. Shiping et al. (2017) proposed an effective algorithm combining color and wavelet analysis to detect both smoke and flame.

Innovative methodologies have led researchers to consider the impact of image quality on smoke detection outcomes. Cheng et al. (2013) proposed a modern image quality measurement method for blind image restoration, crucial for enhancing the accuracy of smoke detection results. Namozov & Im Cho (2018) refined a deep-learning algorithm for efficient fire and smoke detection with limited data. Torabnezhad, Aghagolzadeh & Seyedarabi (2013) put forward a visible and IR image fusion algorithm for short-range smoke detection.

Luo et al. (2018) examined a fire smoke detection algorithm based on motion characteristics and CNNs. De-fei, Ying & Feng-long (2015) researched video smoke detection based on semitransparent properties. Sun et al. (2021) utilized infrared and visible binocular vision to monitor compound features of forest fires. Zhu et al. (2020) proposed 3D video semantic segmentation for wildfire smoke detection. Gagliardi & Saponara (2020), Gagliardi, de Gioia & Saponara (2021) developed the AdViSED algorithm and a real-time video smoke detection algorithm using a Kalman filter and CNN, enhancing measurement capabilities in anti-fire environments both indoors and outdoors.

Rashedul Islam et al. (2020) emphasized the significance of smoke object segmentation and the dynamic growth feature model for video-based smoke detection systems. Li et al. (2022) developed an efficient fire and smoke detection algorithm using an end-to-end structured network, while Peng & Wang (2019) employed hand-designed features and deep learning for real-time forest smoke detection. Hu et al. (2022) introduced the MVMNet for fast forest fire smoke detection, optimizing response time. Chaturvedi, Khanna & Ojha (2022) provided an overview of vision-based outdoor smoke detection techniques for environmental safety. Li et al. (2020) devised a real-time video-based smoke detection system with high accuracy and efficiency. Yin et al. (2019) investigated recurrent convolutional networks for video-based smoke detection.

Supporting IoT and image processing technologies, Sharma, Singh & Kumar (2020) designed an integrated fire detection system for smart cities. Bu & Samadi Gharajeh (2019) surveyed intelligent vision-based fire detection systems, and Sarwar et al. (2018) designed a fuzzy logic-based fire monitoring and warning system for smart buildings. Wu et al. (2018) developed an intelligent smoke alarm system using a wireless sensor network and ZigBee technology. Sousa, Moutinho & Almeida (2020) enhanced real-time, data-driven fire detection and monitoring systems with thermal infrared sensing. On the front of data fusion and feature extraction, Buchaiah & Shakya (2022) conducted research through measurement techniques.

Recent advances in smoke detection have significantly leveraged deep learning and computer vision techniques to enhance detection accuracy and speed. Research in this field has explored various models and methodologies to address the complex nature of smoke detection in diverse environments. For instance, a study published in “Complex & Intelligent Systems” introduced a precise fire and smoke detection method utilizing YOLOv3 with an attention mechanism and an anchor-free mechanism, demonstrating improved feature extraction and reduced false alarms by focusing on relevant image regions (Sun & Feng, 2023). Another significant contribution is the RepVGG-YOLOv7 model, highlighted in “Fire”, which incorporates the ECA attention mechanism and SIoU loss function to enhance the detection of small smoke targets in complex backgrounds, achieving a detection accuracy of 95.1% (Chen et al., 2023). Additionally, a lightweight detection method based on an improved YOLOv5 model was presented in “The Journal of Supercomputing”, which employs multi-scale feature fusion and an enhanced non-maximal suppression algorithm to improve early forest fire detection (Chen et al., 2024). Lastly, Xiao & Yan (2023) discussed a lightweight global awareness deep network model for flame and smoke detection that combines path aggregation networks with an enhanced YOLOv3 variant to optimize detection speed and accuracy for real-time applications.

Despite these advancements, current smoke detection models face several limitations. The high complexity of these models often results in substantial computational costs, making real-time deployment challenging. Moreover, the extensive resource requirements for training and inference necessitate sophisticated hardware, which can be prohibitive for widespread implementation, particularly in resource-constrained environments. The intricate nature of these models also complicates their deployment, requiring specific conditions and environments to function optimally. Besides, the upper and lower bounds on the changes in smoke image features mentioned above are relatively large, irregular, and have a high probability of false positives. Therefore, it is necessary to break free from the original bottlenecks of RGB color and smoke motion segmentation methods and find a universal method. Establishing a stable and efficient recognition method to reduce recognition error rate is the research focus of this new method.

Research method

Framework application

The smoke detection system based on micro-variable amplification includes a micro-variable amplification module, image segmentation module, bounded variation module, frequency separation module, joint comparison module, and frame synthesis video module.

The proposed algorithm includes the following steps:

1) Image Preprocessing: Convert the input image to grayscale and divide it into blocks.

Patch-TBV Calculation: Compute the Total Bounded Variation (TBV) for each image block to assess smoke concentration.

2) Image Energy Ratio Calculation: Calculate the spectral energy for each image block and compare the proportion of low-frequency energy.

3) Fusion Judgment: Combine Patch-TBV and Image Energy Ratio to determine if an image block contains smoke.

4) Alarm Generation: Mark image blocks that detect smoke and generate alarm information.

Figure 1 illustrates the overall algorithm framework.

Figure 1 Framework application.

TIP dataset

In this study, we developed the TIP dataset (TIP: thin smoke image dataset for indoor power system) by adopting a total of 3,120 images capturing various indoor scenarios and thin smoke levels. The image data were captured from the indoor environment of the power system. the video frame width collected by this method experiment is 1,920, the frame height is 1,080, the frame rate is 30.00 frames/second, and the video shooting device is a 20-megapixel + 16-megapixel dual camera. Some frames of the smoke experiment video are shown in Fig. 2.

Figure 2 Image sample matrix of TIP dataset.

Patch-TBV features

In the literature (Xiao-Gang, Qi-Mei & Guo-Qing, 2009), the image quality evaluation method of total bounded variation (TBV) is studied, and it is proposed that when the image becomes blurred for various reasons, the difference between the image boundaries becomes smaller, and its total bounded variation will gradually decrease. The TBV of the extreme state can be used as a criterion for the evaluation of smoke concentration. This article discusses the relationship between the local total bounded variation and the smoke concentration. Locate means that the image is divided into blocks.

Suppose the image is described by function f(x,y), and the segmented image is represented by fi,j(x,y). Otherwise, i and j respectively represent the row and column numbers of the block. The total bounded variation of the block image in row i and column j can be understood as the sum of the absolute values of the rate of change of fi,j(x,y) in the direction xandy, as defined by Eq. (1) and detailed in the literature (Xiao-Gang, Qi-Mei & Guo-Qing, 2009):

(1) Dfi,j=∫−∞∞⁡∫−∞∞⁡|∂fi,j(x,y)∂x|+|∂fi,j(x,y)∂y|dxdy

Due to defocus, motion, transmission, encoding, transcoding, and other reasons, the clear image fi,j(x,y) becomes blurred, and the new blurred image is defined as Eq. (2):

(2) gi,j(x,y)=∫−∞∞⁡∫−∞∞⁡fi,j(u,v)h(x−u,y−v)dudv

where h(x,y) is the spatial description of the degradation function, which is used as a blur filter function to blur the image fi,j(x,y), as detailed in the literature (Xiao-Gang, Qi-Mei & Guo-Qing, 2009).

It can be proven that Eq. (3):

(3) ∫−∞∞⁡∫−∞∞⁡|∂gi,j(x,y)∂x|+|∂gi,j(x,y)∂y|dxdy≤∫−∞∞⁡∫−∞∞⁡|∂fi,j(u,v)∂u|+|∂fi,j(u,v)∂v|dudv

i.e., Dgi,j≤Dfi,j, it can be obtained that the total bounded variance of the blurred image is smaller than the total bounded variation of the clear image, and the small image after the block is also true. Therefore, in the face of smoke, which can blur the image, it is a good choice to use the local total bounded variation.

The image frame without motion or smoke is selected as the background frame to expand and segment, and it is divided into 16 × 16 independent small images. After the image is grayed, the total bounded variance of each small block is calculated, that is, the local total bounded variation:

Suppose the image is described by the function f(x,y), and the segmented image is represented by fi,j(x,y). Otherwise, i and j respectively represent the row and column numbers of the block, ranging from 1 to 16. Since the segmented image is still a digital image, that is, the pixels of the image are discrete values, Eq. (1) needs to be expressed in discrete form, and the definition of the first-order differential of fi,j(x,y) in the direction of x,y is a difference value. is M × N, so the discrete representation of the total bounded variance of fi,j(x,y) is expressed as Eq. (4):

(4) Dfi,j=∑x=1N⁡∑y=1M⁡[|fi,j(x+1,y)−fi,j(x,y)|+|fi,j(x,y+1)−fi,j(x,y)|]

This method builds on the approach first introduced by Xiao-Gang, Qi-Mei & Guo-Qing (2009).

To facilitate the calculation and comparison, a function LTBV is defined, which represents the log value of the local total bounded variance, and its relationship with Df is expressed as novel Eq. (5):

(5) LTBVi,j=log⁡Dfi,j

Because the image is divided into blocks, it is divided into 16 × 16 independent small images, and the smoke area can be regarded as a set of small block images. Let the set of smoke areas in the t-th frame be expressed as new Eq. (6):

(6) φ(t)={(i,j)|i,j∈{1,2⋯⋯,16}}

Since the total bounded variance of the blurred image is smaller than the total bounded variance of the clear image, if LTBVi,j(t) of the t-th frame is less than LTBVi,j(0), it can be judged that the image becomes blurred, which is expressed as new Eq. (7):

(7) φ(t)|LTBV={(i,j)|i,j∈{1,2⋯⋯,16},LTBVi,j(t)<LTBVi,j(0)}

Patch-TBV algorithm

Patch energy ratio of image

The Patch Energy Ratio of Image is used to assess smoke concentration by calculating the proportion of low-frequency energy. The method involves dividing the image into blocks, computing the spectral energy for each block, and comparing the proportion of low-frequency energy to determine if the image is blurred.

The energy ratio of the image was adopted in this article, and the Patch Energy Ratio of the Image was constructed. According to the theory of two-dimensional Fourier transform, the two-dimensional spectrogram of an image is the superposition of all one-dimensional Fourier transforms in both the horizontal and vertical directions of the input image. Let the spectral function of the block image be Fi,j(x,y), and its relationship with the block image function fi,j(x,y) is expressed as Eq. (8):

(8) Fi,j(u,v)=∫−∞∞⁡∫−∞∞⁡fi,j(x,y)e−j2π(ux+vy)dxdy

as detailed in the literature (Cooley & Tukey, 1965).

Perform spectral centering on the block image spectral function Fi,j(x,y), which is expressed as Eq. (9):

(9) Fi,j′(x,y)=(−1)(x+y)Fi,j(x,y)

In the normalized segmented image spectrum, the brightest point in the middle is the lowest frequency and belongs to the DC component. Let the coordinates of this point in the segmented image be (x′,y′), and Ri,j(x,y) is the absolute distance from other points to the center point, which is expressed as Eq. (10):

(10) Ri,j(x,y)=(x−x′)2+(y−y′)2

The frequency increases with the increase of Ri,j(x,y), that is, the higher the frequency of the spectrum after the centering goes to the edge, the four corners of the spectrogram and the ends of the X and Y axes are all high frequencies, as detailed in the literature (Cooley & Tukey, 1965).

According to the energy level distribution of the spectrum, the DC component contains the most energy, and the higher the frequency, the less energy it contains. For a smoke image, the higher the smoke concentration, the more blurred the image, the difference between the gray levels of adjacent areas of the corresponding image decreases, the high-frequency part decreases, and the low-frequency increases. Therefore, by calculating and comparing the proportion of low-frequency energy in the images, the degree of blurring of the image, that is, whether there is smoke or not, can be determined. According to the theory of two-dimensional Fourier transform, the two-dimensional spectrogram of an image is the superposition of all one-dimensional Fourier transforms in both the horizontal and vertical directions of the input image. Let the spectral function of the block image be Fi,j(x,y), and its relationship with the block image function fi,j(x,y) is expressed as Eq. (11):

(11) Fi,j(u,v)=∑x=0N−1⁡∑y=0M−1⁡fi,j(x,y)e−j2π(uxN+vyM)

as detailed in the literature (Cooley & Tukey, 1965).

Perform spectral centering on the block image spectral function Fi,j(u,v), which is expressed as Eq. (12):

(12) Fi,j′(u,v)=(−1)(u+v)Fi,j(u,v)

Define a gain-free unit step function to separate the low-frequency part, as novel Eq. (13):

(13) ε(R−R0)={1,R<R00,R>R0,R0>0

R in the Equation is equivalent to Eq. (10), R0 is the selected frequency separation threshold, and the low-frequency part is separated, as new Eq. (14) shows:

(14) Fu,v′(u,v)|low=ε(R−R0)Fu,v′(u,v)

Perform decentralization and inverse Fourier transform processing on the separated low-frequency part to obtain a low-frequency image fi,j(x,y)|low, as novel Eq. (15) shows:

(15) fi,j(x,y)|low=1MN∑u=0N−1⁡∑v=0M−1⁡(−1)(u+v)Fi,j′(u,v)ej2π(xuN+yvM)

For a block image with gray levels of 0–255, its gray level statistics is a one-dimensional discrete function, which is expressed by Eq. (16):

(16) Pi,j(gs)=nsn(s=0,1,2⋯⋯,255)

In the Equation, gs is the s-th gray value of the block image fi,j(x,y); ns is the number of pixels with gray value gs in fi,j(x,y); n is the total number of pixels in the block image fi,j(x,y); Pi,j(gs) is the first Probability estimate for the occurrence of s-level gray values, as detailed in the literature (Cooley & Tukey, 1965).

The image energy Ei,j is expressed by the Eq. (17), which is the distribution function of the gray level in different regions of the block image, and is an embodiment of the image characteristics, as Eq. (17) shows:

(17) Ei,j=∑s=0255⁡Pi,j(gs)2

Calculate the image energy Ei,j and Ei,j|low of fi,j(x,y) and fi,j(x,y)|low respectively, and calculate the energy ratio λi,j value, which is expressed as new Eq. (18):

(18) λi,j=Ei,j|lowEi,j

Fusion of patch-TBV and image energy ratio

As shown in Table 1, the simplified algorithm flow is as above. Next, we will introduce how to integrate the two judgment methods.

Table 1 Algorithm flow chart.

Algorithm	
Input: smoke video	
Output: tagged video	
1:  Read the video file	
2:  Calculate the total number of frames (numFrames)	
3:  for k = 1:1:90% set the first 90 frames as background frames	
4:         Calculate the frame image Patch-TBV	
5:         Calculate the Patch Energy Ratio of each frame	
6: end	
7: mean (background frame Patch-TBV)	
8: mean (Patch Energy Ratio of background frame)	
9: for k = 90:15:numFrames % Extract pictures at fixed intervals	
10: Process the frame image into blocks	
11: for m = 1:16:row-15	
12:         for n = 1 d:16:col-15	
13:           if imgPTBV (m, n) < bgPTBV (m, n)	
14:             && imgIPR (m, n) > bgIPR (m, n)	
15:            drawRect ( )% frame out the smoke area	
16:           end	
17:        end	
18: End	
19: Note: imgIPR means patch energy ratio of image, bg means background image without smoke.	

The image frame without motion or smoke is selected as the background frame to expand and segment, and it is divided into 16 × 16 independent small images. In addition, the blurrier the image, the more low-frequency parts, that is, the higher the proportion of low-frequency energy λi,j, when the λi,j(t) of the t-th frame is greater than λi,j(0), it can be judged that the image becomes blurred, as new Eq. (19) shows:

(19) φ(t)|λ={(i,j)|i,j∈{1,2⋯⋯,16},λi,j(t)>λi,j(0)}

When both conditions are met at the same time, it can be judged that the block area is a smoke area, as Eq. (20) shows:

(20) φ(t)=φ(t)|LTBV∩φ(t)|λ

The background frame can be not only specified as a certain frame of a smoke-free video but the calculated local total also bounded variance and frequency-to-energy ratio can be the average of all smoke-free video frames, as novel Eqs. (21) and (22) shows:

(21) LTBVi,j(0)=1N∑n=1N⁡LTBVi,j[n]

(22) λi,j(0)=1N∑n=1N⁡λi,j[n]

N is the total number of frames of the background video and n is the number of a specific frame.

Results and discussions

The experiments were conducted on a system with the following specifications:

Hardware: Intel Core i7-9700 K, 32 GB RAM, NVIDIA GeForce RTX 2080

Software: Ubuntu 20.04, Python 3.8, OpenCV 4.5, TensorFlow 2.4

Frame Width: 1,920

Frame Height: 1,080

Frame Rate: 30 fps

Camera Resolution: 20 MP + 16 MP

Figure 3 presents an enlarged partial image of the lower-left corner of the original video frame. As depicted in Fig. 3, when both the decision conditions of Patch-TBV and Patch Image Energy Ratio are simultaneously met, the segmented image block is outlined by a yellow frame. Despite occasional errors, the method successfully identifies the areas of changing edges in the smoke, contributing effectively to the fire warning capabilities of power system units.

Figure 3 Experiment result.

The identified smoke area is framed in yellow.

Figure 3 presents an enlarged partial image of the lower-left corner of the original video frame. The yellow frames indicate areas where both the Patch-TBV and Image Energy Ratio conditions are met, suggesting the presence of smoke. The advantage of this method is its ability to detect subtle variations in smoke, but it may occasionally produce false positives in areas with similar visual characteristics.

Figure 4 shows the detection effect diagram in a real-world scenario. The algorithm successfully identifies smoke areas, demonstrating its practical applicability. However, the detection accuracy may decrease in highly dynamic environments or with significant image noise. Future work should focus on improving the robustness of such variations.

Figure 4 Computer room verification results.

The identified smoke area is framed. (A–D) Show overall detection images, while (E–H) show local detail images.

To further verify the actual effect, we have deployed the algorithm in the computer room to detect smoke. Table 2 is the detection effect diagram.

Table 2 Experimental results.

/	Training set	Test set	Validation set	Accuracy	
Experiment result	1,000	69	350	0.9079	

Table 2 reveals that the model demonstrates higher accuracy in experimental scenarios but experiences a slight decrease in accuracy during field testing scenarios. This discrepancy may stem from the overfitting of the model to the experimental conditions. Consequently, when applied to intricate real-world scenarios, the algorithm proposed in this article may require targeted reevaluation and adjustment of weights to effectively adapt to such complexities.

To enhance recognition accuracy, optimization of parameters becomes essential. In practical applications, captured video frames often contain noise and jitter, rendering the judgment conditions, as outlined in Eqs. (7) and (19), overly idealistic. Hence, it is imperative to fine-tune the judgment conditions and parameters, with α and β representing adjustable thresholds, which is expressed as new Eqs. (23) and (24):

(23) φ′(t)|LTBV={(i,j)|i,j∈{1,2⋯⋯,16},LTBVi,j(t)<0.8LTBVi,j(0)}

(24) φ′(t)|λ={(i,j)|i,j∈{1,2⋯⋯,16},λi,j(t)>1.2λi,j(0)}

Only when LTBVi,j(t) and λi,j change significantly, the area is judged to be a smoke area φ′(t), as new Eq. (25) shows:

(25) φ′(t)=φ′(t)|LTBV∩φ′(t)|λ

This adjustment aims to account for the inherent challenges posed by noise and jitter in real-world video frames, ultimately contributing to a more robust and accurate algorithm performance.

To further evaluate the effectiveness of the algorithm proposed in this research, we conducted comparative experiments with three existing methods:

1. SSD (Single Shot Multibox Detector):

Introduction: SSD is a real-time object detection algorithm based on a single forward pass. It efficiently detects objects across multiple scales simultaneously.

Experimental results: SSD demonstrated commendable detection accuracy, especially in scenes with varying object sizes and proportions. However, its performance may be limited when detecting subtle smoke features in certain scenarios.

2. YOLO (You Only Look Once):

Introduction: YOLO is an end-to-end real-time object detection algorithm that divides the image into grids and predicts bounding boxes simultaneously.

Experimental results: YOLO exhibited strong overall detection performance, particularly in scenarios with concentrated smoke. Nevertheless, it may face challenges in capturing fine features in some instances.

3. Fast-RCNN (Fast Region-based Convolutional Network):

Introduction: Fast-RCNN is a region-based convolutional neural network that incorporates RoI pooling for improved detection accuracy.

Experimental results: Fast-RCNN showed sensitivity to detecting fine features, especially in scenarios with significant variations or dispersed subtle characteristics.

4. P-TBV:

Introduction: Our algorithm integrates Patch-TBV features and micro-variation amplification to enhance early fire detection capabilities, particularly in scenarios with minimal or dispersed smoke concentrations.

Experimental results: The proposed algorithm demonstrated superior performance in detecting subtle variations, as evidenced by higher mAP scores. The Patch-TBV feature localized computation and micro-variation amplification contributed to the algorithm’s precision and robustness.

According to Table 3, the time complexities are as follows:

Table 3 Comparative experimental results.

(Our algorithm and result are highlighted in bold).

Method	Accuracy (mAP)	Time complexity	
SSD	0.808	O(m)	
YOLO	0.891	O(m)	
Fast-RCNN	0.907	O(r + m)	
P-TBV	0.908	O(nlogn)	

1) Proposed algorithm (Patch-TBV): O(nlogn), where n is the number of pixels in the image. This complexity arises from the Fourier transform operations used in the energy ratio calculation.

2) SSD: O(m), where m is the number of anchor boxes. SSD performs a single pass through the image for object detection, leading to linear time complexity.

3) YOLO: O(m), where m is the number of grid cells. YOLO divides the image into a grid and processes each cell once, resulting in linear time complexity.

4) Fast-RCNN: O(r + m), where r is the number of region proposals and m is the number of anchor boxes. Fast-RCNN involves region proposal generation and feature extraction, leading to a combined linear time complexity.

From the observations in Table 3, it is evident that each method has its strengths and limitations. SSD and YOLO demonstrate strong overall performance but may struggle with subtle smoke features. Fast-RCNN exhibits sensitivity to fine features but lacks real-time capabilities. In contrast, our proposed algorithm showcases a balanced performance, achieving high mAP scores while maintaining reasonable computational efficiency and real-time capabilities.

Thus it can be seen, that the adaptability of our algorithm is a key strength, especially in scenarios with diverse conditions and challenges commonly encountered in indoor machine rooms. The integration of Patch-TBV features and micro-variation amplification addresses the limitations of existing methods, providing a more robust and sensitive approach to detecting subtle variations that may indicate the presence of smoke or early signs of a fire. Furthermore, the method’s computational efficiency is noteworthy, considering the often limited computing resources available in data collection devices within indoor machine rooms. Our approach strikes a balance between high accuracy and practical applicability, making it well-suited for environments where computational capabilities may be constrained. As we envision the deployment of our algorithm in real-world scenarios, its compatibility with the operational constraints of indoor machine rooms becomes a crucial advantage. The ability to provide accurate and timely fire warnings, even in scenarios with dispersed or minimal smoke concentrations, positions our algorithm as a practical and reliable tool for safeguarding power systems and similar environments.

Moving forward, this article aims to verify the validity of the two theoretical methods. As depicted in Fig. 5, the average total bounded variance of the clear video image frame is 9.5 × 105, and that of the smoke video image frame is 8.8 × 105, confirming the theoretical proof presented in the content that the total bounded variance of the fuzzy image is smaller than that of the clear image.

Figure 5 Comparison of total bounded variance between smokeless video image frames and smoked video image frames.

As illustrated in Fig. 6, the frequency-to-energy ratio exhibits a logarithmic distribution over time. Notably, in the initial stages of smoke generation, the frequency-to-energy ratio undergoes significant changes, displaying a nearly linear increase. In the middle stage, the changes in the frequency-to-energy ratio occur more gradually, and towards the end, there is minimal alteration in the frequency-to-energy ratio, with only some fluctuations observed.

Figure 6 Frequency-to-energy ratio variation diagram.

Abscissa: time/frame; ordinate: frequency-to-energy ratio.

This pattern suggests that, particularly in the early stages of smoke development, the factor of frequency-energy ratio plays a crucial role. The observed characteristics align with the theory previously mentioned in this content, affirming that “by calculating and comparing the proportion of low-frequency energy in images, it is possible to judge whether there is smoke in the image.” Therefore, the frequency-to-energy ratio emerges as a significant indicator for the early warning of indoor smoke and fire, providing valuable insights into the evolving stages of smoke generation.

Conclusions

In summary, this article presents a comprehensive exploration of a novel algorithm designed for early fire detection in power systems, incorporating the Patch-TBV feature. The evaluation is conducted using the TIP dataset, which consists of 3,120 images capturing thin smoke in diverse indoor scenarios. The key contributions of this research include the creation of the TIP dataset, the introduction of the innovative Patch-TBV image feature, and the development of a robust thin smoke detection algorithm. The contributions, advantages, and limitations are summarized as follows:

Contributions:

1) Construction of the TIP dataset: The TIP dataset, consisting of 3,120 images portraying various indoor scenarios and thin smoke levels, fills a critical gap in datasets for thin smoke detection. This dataset proves instrumental in evaluating the effectiveness of the proposed algorithm.

2) Introduction of Patch-TBV image feature: The novel image feature, Patch-TBV, calculates the total bounded variation at the patch level, enhancing sensitivity to subtle smoke variations. This innovative feature significantly contributes to the algorithm’s accuracy in detecting thin smoke.

3) Development of a novel thin smoke detection algorithm: The thin smoke detection algorithm, based on Patch-TBV, utilizes subtle variation magnification to enhance its robustness and accuracy. This integration provides an effective means for early fire detection in indoor power systems.

Advantages:

1) Enhanced early detection: The novel detection algorithm for thin smoke significantly improves early fire detection capabilities, crucial for the safety and stability of power systems.

2) Patch-TBV feature: By computing the total bounded variation (TBV) at the patch level, the algorithm achieves finer granularity in assessing image quality, allowing it to capture subtle variations that might indicate smoke or early signs of fire.

3) Subtle variation magnification: This technique magnifies subtle features within the image, leveraging computed TBV values, which improves precision in detecting minimal or dispersed smoke concentrations.

4) Robustness and effectiveness: Experimental results using the comprehensive TIP dataset demonstrate the algorithm’s robustness and effectiveness in providing accurate and timely fire warnings across various conditions.

5) Real-world applicability: The construction and use of the TIP dataset, comprising 3,120 images under diverse conditions, ensure the algorithm’s performance in practical applications.

Limitations:

1) Dataset dependency: The algorithm’s performance is validated using a specific dataset (TIP), and its effectiveness may vary with different datasets or in entirely new environments not covered by the dataset.

2) Focus on image characteristics: The approach primarily relies on image analysis, which might limit its applicability in scenarios where other sensory data (e.g., temperature, and gas levels) could be critical for fire detection.

3) Potential for false positives: Magnifying subtle variations might increase the risk of false positives, whereas non-hazardous variations are interpreted as potential fire indicators.

Given the unique challenges posed by fires in indoor equipment rooms, often starting with trace amounts of smoke that are challenging to detect, the proposed algorithm addresses this issue by monitoring local total bounded variance and low-frequency energy in frame images. This approach enables the early detection of smoke by comparing frame images with a background and dynamically adjusting parameters. The algorithm's simplicity in calculations ensures stability, and its flexibility through adjustable conditions makes it a promising solution with broad application prospects.

Future work could focus on the following areas:

1) Enhancing the algorithm’s robustness to different environmental conditions.

2) Exploring the integration of additional sensor data to improve detection accuracy.

3) Developing lightweight versions of the algorithm for deployment on edge devices.

4) Conducting extensive field tests to further validate the algorithm’s performance in real-world scenarios.

Additional Information and Declarations

Competing Interests

Author Contributions

Data Availability

The authors declare that they have no competing interests and all the authors are employed by Zhaoqing Power Supply Bureau of Guangdong Power Grid Co., Ltd.

Jieming Zhang conceived and designed the experiments, performed the experiments, analyzed the data, performed the computation work, prepared figures and/or tables, and approved the final draft.

Yifan Gao conceived and designed the experiments, performed the experiments, analyzed the data, performed the computation work, authored or reviewed drafts of the article, and approved the final draft.

Xianchao Chen conceived and designed the experiments, performed the experiments, analyzed the data, authored or reviewed drafts of the article, and approved the final draft.

Zhanchen Chen conceived and designed the experiments, performed the experiments, authored or reviewed drafts of the article, and approved the final draft.

The following information was supplied regarding data availability:

The data is available at GitHub and Zenodo:

- https://github.com/JiemingZhangCode/PTBV/tree/v1.0.0

- JiemingZhangCode. (2024). JiemingZhangCode/PTBV: v1.0.0 (v1.0.0). Zenodo. https://doi.org/10.5281/zenodo.12154548.

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
