# Peer review of "A flexible perception method of thin smoke based on patch total bounded variation for buildings"

_PeerJ Computer Science, doi:10.7717/peerj-cs.2282_

## Round 0.1 · original submission · Major Revisions

· Academic Editor

Major Revisions

Dear authors,

Thank you for submitting your article. Reviewers have now commented on your article and suggest major revisions. When submitting the revised version of your article, it will be better to also address the following:

1. The research gaps and contributions should be clearly summarized in the introduction section. Please evaluate how your study is different from others in the related work section.
2. The paper lacks the running environment, including software and hardware. The analysis and configurations of experiments should be presented in detail for reproducibility. It is convenient for other researchers to redo your experiments and this makes your work easy acceptance. A table with parameter settings for experimental results and analysis should be included in order to clearly describe them. The values for the parameters of the algorithms should be given.
3. The authors should clarify the pros and cons of the methods. What are the limitation(s) methodology(ies) adopted in this work? Please indicate practical advantages, and discuss research limitations.
4. Explanation of the equations should be checked. All variables should be written in italic as in the equations.
5. Please use equation numbers for referencing the equations. Do not use “following” “as follows”, etc. Furthermore, appropriate references should be used for relevant equations. They seem they are firstly used in this paper.
6. Please include future research directions.

Reviewer 1 ·

Basic reporting

The analysis of the current research status in this field in the Introduction is not sufficient, and the description of the problems to be solved should be more clearly.
Some descriptions in the Patch Energy Ratio of Image are too detailed, and general processing suggestions are simplified.
The full text should provide a more detailed description of the algorithm proposed in this article, lacking a detailed description of the overall algorithm. The algorithm process described in Figure 1 is too simple.

Experimental design

The experimental validation section was validated using actual engineering datasets, which lacked persuasiveness. It is recommended that the author add a universal dataset for validation to further validate the effectiveness.

Validity of the findings

The experimental section only provided the overall results, lacking validation through ablation experiments.

Additional comments

1. The analysis of the current research status in this field in the Introduction is not sufficient, and the description of the problems to be solved should be more clearly.

2. Some descriptions in the Patch Energy Ratio of Image are too detailed, and general processing suggestions are simplified.

3. The full text should provide a more detailed description of the algorithm proposed in this article, lacking a detailed description of the overall algorithm. The algorithm process described in Figure 1 is too simple.

4. The experimental validation section was validated using actual engineering datasets, which lacked persuasiveness. It is recommended that the author add a universal dataset for validation to further validate the effectiveness.

5. The experimental section only provided the overall results, lacking validation through ablation experiments.

Reviewer 2 ·

Basic reporting

In this study, a novel detection algorithm for thin smoke was proposed to enhance early fire detection capabilities. The core is that the Patch-TBV feature was proposed first, and the total bounded variation (TBV) was computed at the patch level. This approach is rooted in the understanding that traditional methods struggle to detect minute variations in image characteristics, particularly in scenarios where the features are dispersed or subtle. By computing TBV at a more localized level, the algorithm proposed gains a finer granularity in assessing image quality, enabling it to capture subtle variations that might indicate the presence of smoke or early signs of a fire.

There are some points that need further clarification and improvement:
The methodology lacks sufficient detail. Specifics regarding dataset creation, pre-processing steps, model architectures, and hyperparameter tuning are crucial for reproducibility and understanding the study's validity.
The choice of models seems arbitrary. There is no clear justification for why these models were selected over others or how they are suited to the fire detection task.

Experimental design

The evaluation lacks rigor and depth. The reported accuracy rates seem inflated, especially without proper validation techniques such as cross-validation or testing on an independent dataset.
The dataset provided on github contains very limited images (burning cigarettes), it needs to be increased.
The number of data used for testing is too small for generalization.
The time complexity should also be given in Table 3.

Validity of the findings

The discussion section lacks depth and fails to provide meaningful insights into the results. There is a need for a more thorough analysis of the findings, including potential limitations and avenues for future research. Figures 3 and 4 should be discussed in more detail with their advantages and disadvantages.
The conclusion is weak and fails to provide meaningful insights or implications for future research. It merely restates the results without discussing potential avenues for improvement or addressing limitations.

Additional comments

In the abstract section, it should start with a capital letter after the point sign (another).
In the research method section, headings 3 and 4 are the same.
Equation 1 and 4 are the same.
Equation 9 and 12 are the same.
Equation 10 and 14 are the same.

---

## Round 0.2 · Minor Revisions

· Academic Editor

Minor Revisions

Dear authors,

Thank you for the revised manuscript. Based on the one reviewer's comments, minor revision is required for the manuscript. When submitting the revised version of your article, it will be better to also address the following:

1. Explanation of the equations should be checked. Please use equation numbers for referencing the equations. Do not use "as", “following” “as follows”, etc. Furthermore, appropriate references should be used for relevant equations. They seem they are firstly used in this paper.
2. Pros and cons of the methods should be clarified. What are the limitation(s) methodology(ies) adopted in this work? Please indicate practical advantages, and discuss research limitations.

Best wishes,

Reviewer 2 ·

Basic reporting

In this study, a novel detection algorithm for thin smoke was proposed to enhance early fire detection capabilities. The core is that the Patch-TBV feature was proposed first, and the total bounded variation (TBV) was computed at the patch level. This approach is rooted in the understanding that traditional methods struggle to detect minute variations in image characteristics, particularly in scenarios where the features are dispersed or subtle. By computing TBV at a more localized level, the algorithm proposed gains a finer granularity in assessing image quality, enabling it to capture subtle variations that might indicate the presence of smoke or early signs of a fire.
Compared to the previous version, the methodology section is better explained.

Experimental design

According to the previous version:
The dataset could not be increased.
The time complexity is given in Table 3.

Validity of the findings

According to the previous version:
Figures 3 and 4 are better discussed.
Conclusion section has been improved.

Additional comments

In the research method section, headings 3 and 4 are the same (3. Patch-TBV features 4. Patch-TBV features)..It should be changed.

---

## Round 0.3 · Minor Revisions

· Academic Editor

Minor Revisions

Dear authors,

The comments on the revised manuscript are included at the end of this letter. We ask that you make changes to your manuscript based on these comments.

Best wishes,

Reviewer 1 ·

Basic reporting

This article proposes the application of "the total bounded variation" for detecting thin smoke and conducts experimental verification. The paper has a certain degree of innovation.The English description should be checked, as some descriptions are not rigorous enough.There are deficiencies in the analysis of the current situation in this field in the Introduction, and the latest relevant literature should be added and analyzed.

Experimental design

The SSD and YOLO algorithms selected in the comparative experiment are object detection algorithms, and the author should choose relevant algorithms in this field for comparison to highlight the effectiveness of this paper.
The author should add ablation experiments to demonstrate the effectiveness of applying this algorithm in this article.

Validity of the findings

no comment

Additional comments

no comment

Reviewer 2 ·

Basic reporting

In this study, a novel detection algorithm for thin smoke was proposed to enhance early fire detection capabilities. The core is that the Patch-TBV feature was proposed first, and the total bounded variation (TBV) was computed at the patch level. This approach is rooted in the understanding that traditional methods struggle to detect minute variations in image characteristics, particularly in scenarios where the features are dispersed or subtle. By computing TBV at a more localized level, the algorithm proposed gains a finer granularity in assessing image quality, enabling it to capture subtle variations that might indicate the presence of smoke or early signs of a fire.
Compared to the previous version, the methodology section is better explained.

Experimental design

According to the previous version:
The dataset could not be increased.
The time complexity is given in Table 3.

Validity of the findings

According to the previous version:
Figures 3 and 4 are better discussed.
Conclusion section has been improved.

---

## Round 0.4 · accepted · Accept

· Academic Editor

Accept

Dear authors,

Thank you for clearly addressing all the reviewers' comments. I confirm that the quality of your paper is improved. The paper now seems to be ready for publication in light of this revision.

Best wishes,

Reviewer 1 ·

Basic reporting

The authors of this article have made revisions in accordance with the review requirements.

Experimental design

The authors of this article have made revisions in accordance with the review requirements.

Validity of the findings

The authors of this article have made revisions in accordance with the review requirements.

Additional comments

The authors of this article have made revisions in accordance with the review requirements.